

# Using the wax moth larva *Galleria mellonella* infection model to detect emerging bacterial pathogens

Rafael J. Hernandez[1,4,*], Elze Hesse[2,*], Andrea J. Dowling[2], Nicola M. Coyle[3], Edward J. Feil[3], Will H. Gaze[4] and Michiel Vos[4]

[1] Stony Brook School of Medicine, Department of Global Medical Education, State University of New York at Stony Brook, Stony Brook, NY, USA
[2] Department of Biosciences, University of Exeter, Penryn, UK
[3] The Milner Centre for Evolution, Department of Biology and Biochemistry, University of Bath, Bath, UK
[4] European Centre for Environment and Human Health, University of Exeter, Penryn, UK
* These authors contributed equally to this work.

## ABSTRACT

Climate change, changing farming practices, social and demographic changes and rising levels of antibiotic resistance are likely to lead to future increases in opportunistic bacterial infections that are more difficult to treat. Uncovering the prevalence and identity of pathogenic bacteria in the environment is key to assessing transmission risks. We describe the first use of the Wax moth larva *Galleria mellonella*, a well-established model for the mammalian innate immune system, to selectively enrich and characterize pathogens from coastal environments in the South West of the UK. Whole-genome sequencing of highly virulent isolates revealed amongst others a *Proteus mirabilis* strain carrying the *Salmonella* SGI1 genomic island not reported from the UK before and the recently described species *Vibrio injenensis* hitherto only reported from human patients in Korea. Our novel method has the power to detect bacterial pathogens in the environment that potentially pose a serious risk to public health.

## INTRODUCTION

Emerging infectious diseases (EIDs) pose a major threat to human health (*Jones et al., 2008*). A large proportion of EIDs are caused by bacteria (estimated to be 54% (*Jones et al., 2008*) and 38% (*Taylor, Latham & Woolhouse, 2001*)). Although most emerging bacterial pathogens have zoonotic origins, a large proportion of infectious bacteria are free-living, for instance being associated with food (*Newell et al., 2010*), drinking water (*Ford, 1999*) or recreational waters (*Baker-Austin et al., 2013*). Microbial safety is routinely assessed through the quantification of faecal indicator bacteria (FIB) (*World Health Organization, 2003*). However, many FIB lineages are not associated with disease and there is no a priori reason to expect a relationship between FIB abundance and

Corresponding author
Michiel Vos, m.vos@exeter.ac.uk

non-gastrointestinal disease (e.g., ear or skin infections). There are dozens of bacterial genera occurring in natural environments that are not primarily associated with human or animal faecal contamination but that are able to cause opportunistic infections (*Berg, Eberl & Hartmann, 2005*). Alternatives to FIB such as quantification of pathogen-specific genes via molecular methods (*Girones et al., 2010*), flow cytometry (*Prest et al., 2013*) or isolation of specific pathogens (*Kaysner et al., 1987*) either are not linked to infection risk, are based on costly methodologies or are limited to a subset of "known knowns." The current lack of a direct screening method for the presence of pathogenic bacteria in environmental samples is therefore a major barrier to understanding drivers of virulence and ultimately infection risk.

We demonstrate the use of the Wax moth larva *Galleria mellonella* as a bioindicator for microbial water quality, and a means to selectively isolate and characterize pathogens. *G. mellonella* is a well-established model system for the mammalian innate immune system and has been used extensively to test for virulence in a range of human pathogens by quantifying survival rate after injection of a defined titre of a specific strain or mutant (*Ramarao, Nielsen-Leroux & Lereclus, 2012*). Bacterial virulence in *Galleria* is positively correlated with virulence in mice (*Brennan et al., 2002*) as well as macrophages (*Wand et al., 2011*). Instead of quantifying the virulence of a specific bacterial clone, here we measure *Galleria* survival after injection with entire microbial communities from concentrated environmental water and sediment-wash samples. We isolate bacteria responsible for *Galleria* mortality and assess their pathogenic potential through whole-genome sequencing.

## MATERIALS AND METHODS

### Sample collection and processing

Eight sampling sites on the Fal estuary and English Channel coast near Falmouth, Cornwall, UK (50°.16′N–5°.10′W) were selected for water and sediment sampling. Four sites were estuarine (Mylor, Flushing, Penryn and Falmouth Harbour) and four truly coastal (Castle Beach, Gyllyngvase Beach, Swanpool Beach and Maenporth Beach). Samples from each site were collected on June 21, 2017 and July 6, 2017. Each sampling effort consisted of collecting water ($2 \times 50$ mL) and the upper one cm layer of sediment (~25 g) from all eight sites within a time span of 2 h around low-tide. Samples were collected using sterile 50 mL centrifuge tubes. All samples were kept on ice during transport and processed in the laboratory within an hour from collection.

Each duplicate water sample was centrifuged at 500 rpm at 4 °C for 15 min to pellet sediment particles. Supernatants were then transferred to sterile 50 mL centrifuge tubes and spun down at 3,500 rpm at 4 °C for 30 min to pellet bacteria. Supernatants were then discarded and the pellet in each tube resuspended in 500 μL of sterile M9 buffer (12.8 g $L^{-1}$ $Na_2HPO_4$-$7H_2O$; 3 g $L^{-1}$ $KH_2PO_4$; 0.5 g $L^{-1}$ NaCl; 0.1 g $L^{-1}$ $NH_4Cl$). Resuspensions from duplicate samples were then combined in a 1.5 mL Eppendorf tube to obtain a total concentrated water sample of one mL. Bacteria were extracted from sediments by adding 10 g of sediment to 10 mL of sterile M9 buffer. Samples were vortexed at 3,000 rpm for 2 min, and then centrifuged at 500 rpm at 4 °C for 15 min to pellet

sediment particles. The supernatant was then centrifuged as above and resuspended in one mL of M9 buffer. All samples were kept at 4 °C or on ice at all times.

## Culturable bacteria

A total of 100 µL per water or sediment-wash sample was plated onto both LB agar (Fisher BioReagents, Loughborough, UK) and coliform agar (Millipore Sigma, Burlington, MA, USA) and incubated overnight at 37 °C. Colony Forming Units (CFU) per plate were counted after 24 h incubation. Colonies on coliform agar were scored as dark-blue ("*Escherichia coli*"), pink ("coliform") or white/transparent ("other"). Units are reported in CFU/100 mL for water samples and CFU/10 g for sediment samples.

## Flow cytometry

Concentrated water and sediment-wash samples were analyzed for bacterial load using flow cytometry. For each sample (200 µL sample + 20 µL 2.5 × SybrGold), 10 µL was analyzed for number of events using low flow rate (14 µL/min) in a BD Accuri C6 Plus flow cytometer (BD Biosciences, San Jose, CA, USA). The BD Accuri C6 Plus is equipped with a blue and red laser, two light scatter detectors and four fluorescence detectors. Bacteria cells were distinguished from non-biological particles using 533/30 filter in FL1 (indicative of nucleic acid content), 670 LP in FL3 (indicative of chlorophyll content) and FSC (indicative of cell size). Data were analyzed using the BD Accuri C6 software. A non-dye sample was included to control for auto-fluorescence.

## *Galleria mellonella* assay for environmental samples

Final instar waxmoth (*G. mellonella*) larvae (~220 mg each) were purchased from Livefood U.K. (http://www.livefood.co.uk), stored in the dark at 4 °C and used within 2 weeks (all experiments utilized larvae from the same batch). A 100 µL Hamilton syringe (Sigma-Aldrich Ltd, Gillingham, UK) with 0.3 × 13 mm needle (BD Microlance 3, Becton Dickinson, Plymouth, UK) was used to inject larvae with 10 µL of sample into the last left proleg. New sterile needles were used for injection of each sample, and the syringe was cleaned between sample injections with 70% ethanol and sterile M9 buffer. All water and sediment washes (eight locations × two timepoints) were assayed in one experiment using 20 *Galleria* per sample. Three negative controls were used for each experiment: a no-injection control (to control for background larvae mortality), a buffer control in which larvae were injected with 10 µL of M9 buffer (to control for impact of physical trauma), and a filter-sterilized sample control in which larvae were injected with filter-sterilized samples (0.22 µm syringe filters; Thermo Fisher, Waltham, MA, USA) to account for any toxic chemicals present in the sample. Before injection, the *Galleria* larvae were separated into Petri dishes in groups of 20 and anesthetized by placing on ice for approximately 30 min before injection. After injection, Petri dishes were incubated at 37 °C and inspected at 24, 48 and 72 h post-injection to record morbidity and mortality. *Galleria* larvae were considered infected if they expressed dark pigmentation (melanisation) after inoculation. Larvae were scored as dead if they did not respond to touch stimuli by blunt sterile forceps.

### *Galleria mellonella* assay for individual clones

To isolate and identify responsible pathogens, a subset of samples showing high *Galleria* mortality were inoculated and incubated again as described above. Eight live larvae showing signs of infection (melanisation) were selected at random from each sample group and dissected for haemocoel collection. Before haemocoel collection, the *Galleria* larvae were anesthetized on ice for 30 min and the site of dissection was sterilized with 70% ethanol. Sterile micro-scissors were used to remove the last left proleg, and a drop of haemocoel was allowed to exude from the larvae before being collected with a pipette. We chose this method over whole-larvae destruction to minimize contamination by commensal gut and skin bacteria. Approximately 5–15 µL of haemocoel was collected per larva and diluted into 500 µL of sterile M9 buffer in an Eppendorf tube. Serial dilutions of each sample were plated onto LB- and coliform agar plates and incubated overnight at 37 °C. A single colony per sample was picked from these plates and grown for 24 h in five mL of LB broth at 37 °C at 180 rpm. Overnight cultures were diluted in broth to a turbidity equivalent to a McFarland standard of 0.5 at 625 nm as measured by spectrophotometry (Bibby Scientific Ltd, Staffordshire, UK). A serial dilution was plated on LB agar and incubated for 24 h at 37 °C to obtain cell densities (CFU/mL). Individual clones were used to inoculate groups of 20 *Galleria* larvae with 10 µL of $1 \times 10^2$ CFU, $1 \times 10^4$ CFU and $1 \times 10^6$ CFU. Larvae were incubated as described above and morbidity and mortality was recorded hourly after an initial 10 h period for a total of 37 h allowing for construction of survival plots and calculation of the 50% lethal dose value ($LD_{50}$).

### Whole-genome sequencing

DNA isolation, Illumina HiSeq sequencing and basic bioinformatics was performed through the MicrobeNG program in Birmingham, U.K. Vials containing beads inoculated with liquid culture were washed with extraction buffer containing lysostaphin and RNase A, incubated for 25 min at 37 °C. Proteinase K and RNaseA were added and incubated for 5 min at 65 °C. Genomic DNA was purified using an equal volume of SPRI beads and resuspended in EB buffer. DNA was quantified in triplicates with the Quantit dsDNA HS assay in an Eppendorff AF2200 plate reader. Genomic DNA libraries were prepared using Nextera XT Library Prep Kit (Illumina, San Diego, CA, USA) following the manufacturer's protocol with the following modifications: two nanograms of DNA instead of one were used as input, and PCR elongation time was increased to 1 min from 30 s. DNA quantification and library preparation were carried out on a Hamilton Microlab STAR automated liquid handling system. Pooled libraries were quantified using the Kapa Biosystems Library Quantification Kit for Illumina on a Roche LightCycler 96 qPCR machine. Libraries were sequenced on the Illumina HiSeq using a 250 bp paired end protocol.

### Bioinformatics

Reads were adapter trimmed using Trimmomatic 0.30 with a sliding window quality cut-off of Q15 (*Bolger, Lohse & Usadel, 2014*). De novo assembly was performed on

samples using SPAdes version 3.7 (*Bankevich et al., 2012*), and contigs were annotated using Prokka 1.11 (*Seemann, 2014*). Kraken was used to indicate the likely species classification of our isolates (*Wood & Salzberg, 2014*). For species with established Multi Locus Sequence Typing (MLST) schemes, the sequence type of each isolate was identified using the Centre for Genomic Epidemiology MLST tool (*Larsen et al., 2012*). Other isolates belonging to the same sequence types were found and accessed using databases Enterobase (*Alikhan et al., 2018*) (*E. coli*) and (*Jolley & Maiden, 2010*) (*Pseudomonas aeruginosa*). Whole-genome alignments were constructed using Progressive Mauve (*Darling, Mau & Perna, 2010*). Percentage similarity was calculated using the number of Single Nucleotide Polymorphisms (SNPs) found between the query isolate and the reference sequence. In the case of *V. injenesis* where a closed reference genome was not available, nucleotide identity was confirmed using a core-genome alignment of a closely related draft genome. Core genomes were constructed using Roary 3.12.0 (*Page et al., 2015*).

Differences in gene content (95% nucleotide identity threshold) between the *V. injenesis* reference strain and the *V. injenesis* clone isolated in this study were also identified using Roary 3.12.0. The presence of PCAMU-SGI1 in the *Proteus mirabilis* isolate was assessed using BLAST (*Altschul et al., 1990*). Assemblies were screened for antimicrobial resistance (AMR) related genes and virulence related genes using ABRicate (https://github.com/tseemann/abricate) using 90% and 75% nucleotide identity, and 80% length identity cut-offs. The CARD database was used to identify AMR genes (*McArthur et al., 2013*), and the VFDB virulence factor database was used to find putative virulence factors (*Chen et al., 2015*). BRIG version 0.95 was used to represent circular draft assemblies and indicate the position of AMR and virulence genes on each contig (*Alikhan et al., 2011*). Contigs were ordered by whole-genome alignment against a reference genome using progressive mauve (*Darling, Mau & Perna, 2010*). The *E. coli* isolate was aligned against K-12 substr. MG1655 (NC_000913.3), the *Proteus mirabilis* isolate was aligned against HI4320 (NC_010554.1) and the *Pseudomonas aeruginosa* against PAO1 (NC_002516.2). The *Vibrio injenensis* genome could not be aligned to a closed reference genome and so the contigs could not be ordered or separated by chromosome.

## RESULTS

Our survival assay shows that *Galleria* mortality after 72 h varied widely between both water and sediment samples collected at two dates from eight locations across Cornwall (UK), ranging from 0% to 95% (Fig. 1). Injection of buffer solution or filtered (0.22 μm) samples yielded zero mortality, demonstrating that injection was not harmful and that samples did not contain lethal concentrations of pollutants or toxins. Mortality was largely congruent with FIB counts as well as total bacteria density (as quantified by flow cytometry and total viable counts on LB, Fig. S1), although there was substantial variation (Fig. S2 and Supplemental Information 1).

We chose four environmental samples exhibiting high (≥70%) *Galleria* mortality to isolate the clone(s) responsible for infection and reinoculated these samples to isolate bacteria from the haemocoel of infected, freshly killed larvae. All samples yielded a single colony type on each agar type, indicating that infections were (largely) clonal.

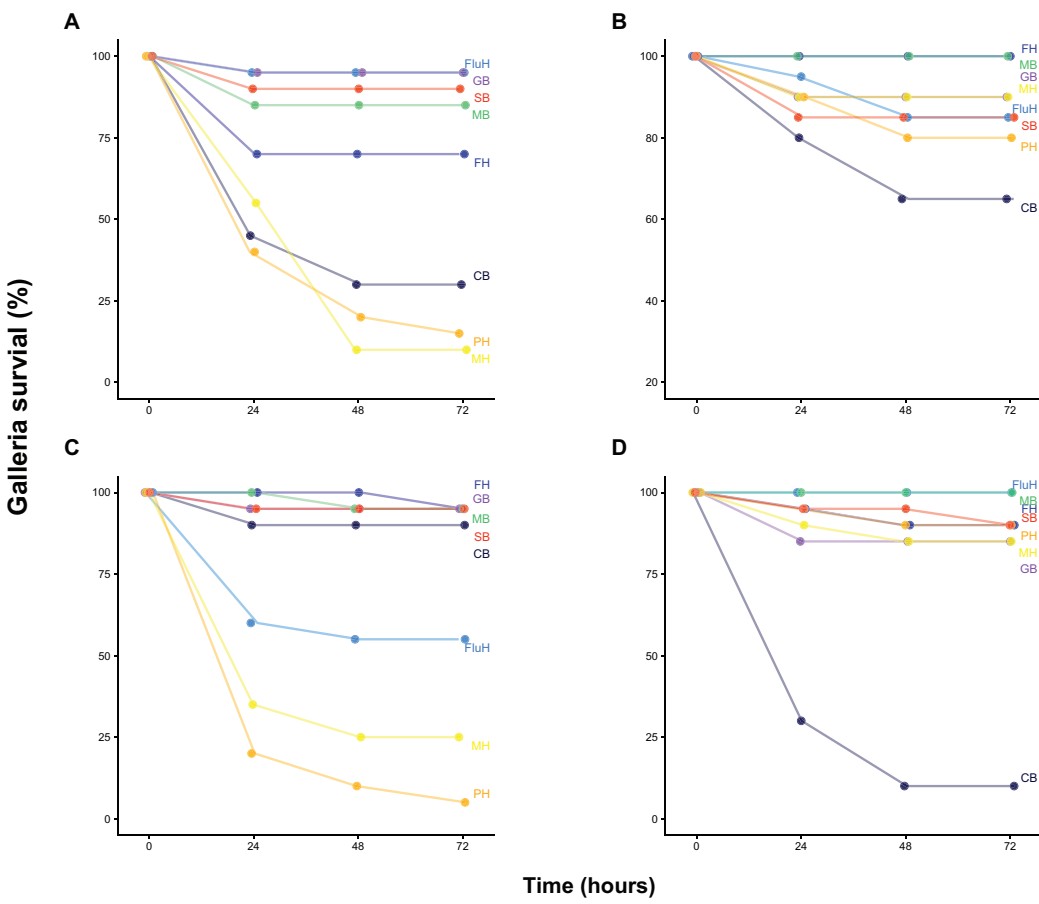

**Figure 1** ***Galleria*** **survival after inoculation with water and sediment samples.** *Galleria* survival (%, 20 individuals per group) was measured 24, 48 and 72 h post-injection with 10 µL of 100-fold concentrated water (B, June 21st, D, July 6th) or sediment wash (see Methods) (A, June 21st, C, July 6th). Sample site abbreviations: CB, Castle Beach; FH, Falmouth Harbour; FluH, Flushing Harbour; GB, Gyllyngvase Beach; MB, Maenporth Beach; MH, Mylor Harbour; PH, Penryn Harbour; SB, Swanpool Beach.

A single clone was picked for each sample, grown up and assayed using three inoculation densities ($1 \times 10^2$ CFU, $1 \times 10^4$ CFU and $1 \times 10^6$ CFU) (Fig. 2). All clones displayed high levels of virulence and were characterized using whole-genome sequencing (Fig. 2). We specifically focused on the identification of virulence- and antibiotic resistance genes (ARGs) as compiled in the VFDB (*Chen et al., 2015*) and CARD (*Jia et al., 2016*) databases, respectively.

The first clone, isolated from estuarine mud (Supplemental Information 1) was identified as the enteric species *Proteus mirabilis*, most closely related to pathogenic strain HI4320 (*Pearson et al., 2008*) (Fig. 2B). Interestingly, this strain was found to carry a multidrug resistance genomic island (SGI1), first identified in an epidemic *Salmonella enterica* serovar Typhimurium clone in the 1990s (*Boyd et al., 2001*). This island has since been found in *Proteus mirabilis* isolated from human patients as well as from animals (*Siebor & Neuwirth, 2013*) but to our knowledge not from *Proteus* strains isolated from natural environments. No virulence genes were found using a 90% similarity cut-off,

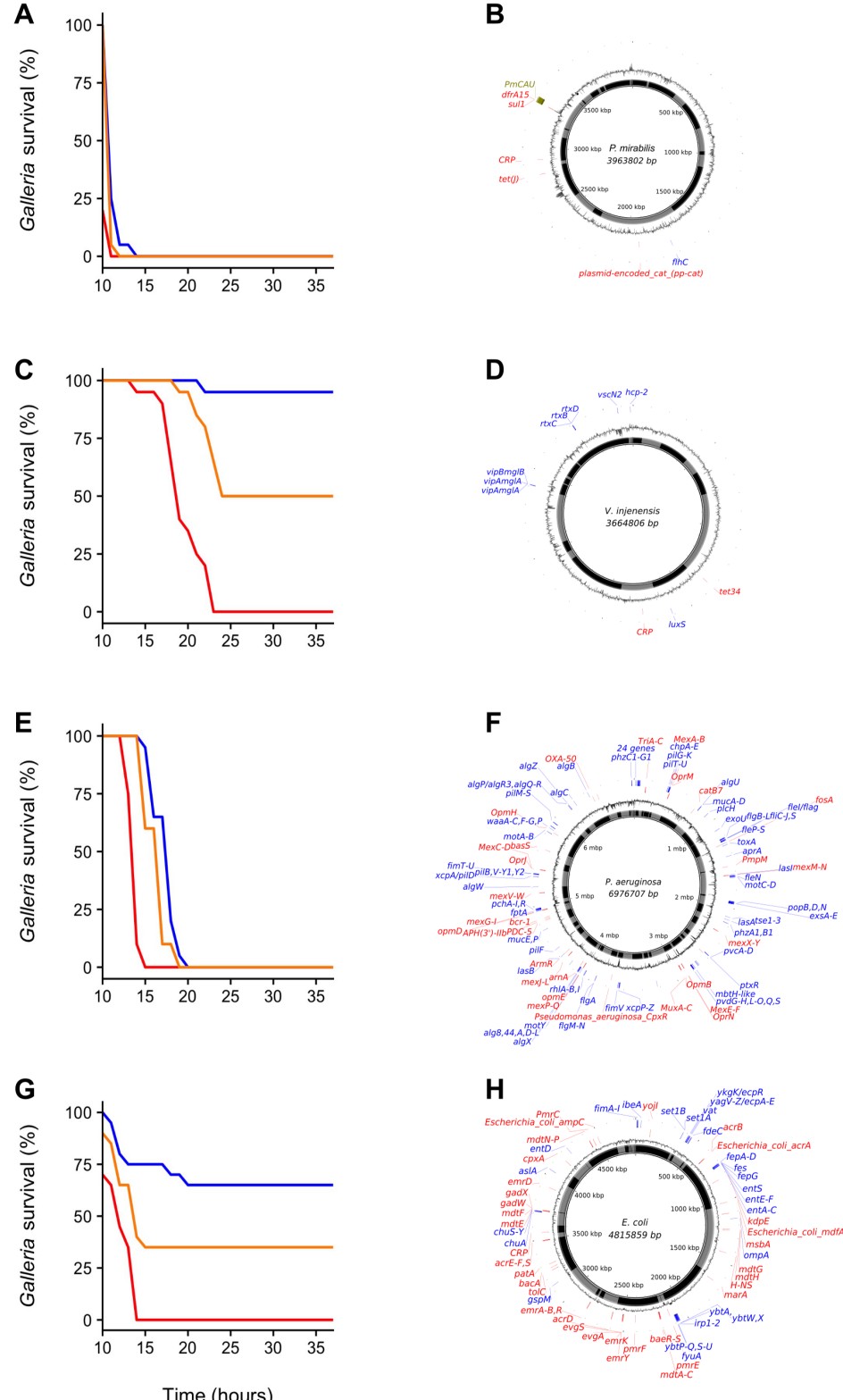

**Figure 2** *Galleria* **survival curves and whole-genome sequence representations of four pathogenic bacterial isolates.** (A, C, E and G) *Galleria mellonella* mortality after inoculation with individual bacterial clones originally isolated from *G. mellonella* larvae infected with environmental (whole-bacterial

**Figure 2** (continued)
community) samples. Groups of 20 *Galleria* larvae were inoculated with 10 µL of $1 \times 10^2$ CFU (blue),
$1 \times 10^4$ CFU (orange) and $1 \times 10^6$ CFU (red). (B, D, F and H) show clone-genome information
(species name and genome size (middle), contigs (inner ring; gray and black), GC content (outer
ring), virulence genes (blue) and ARGs (red) ($\geq 75\%$ nucleotide similarity used for *Proteus mirabilis*
and *V. injenesis*; $\geq 90\%$ similarity used for *Pseudomonas aeruginosa* and *E. coli*; $\geq 80\%$ coverage
criterion for all four species). *Galleria mellonella* mortality after inoculation with *Proteus mirabilis*
($LD_{50} = 1 \times 10^2$ CFU) (A) and its genome (the genomic island SGI1-PmCAU is indicated in green)
(B); *Galleria mellonella* mortality after inoculation with *Vibrio injenensis* ($LD_{50} = 1 \times 10^6$ CFU)
(C) and its genome (note that the absence of a closed draft genome means that contigs are randomly
ordered) (D); *Galleria mellonella* mortality after inoculation with *Pseudomonas aeruginosa* ($LD_{50} =
1 \times 10^2$ CFU) (E) and its genome (F); *Galleria mellonella* mortality after inoculation with *Escherichia
coli* ($LD_{50} = 1 \times 10^4$ CFU) (G) and its genome (H).

but several were identified using a 75% cut-off (Table S2). The clone contains several
ARGs, including the tetracycline efflux protein *TetJ* and *AAC(6′)-Ib7*, a plasmid-encoded
aminoglycoside acetyltransferase (90% similarity cut-off, Table S3).

The second clone, isolated from beach sand, was found to belong to *Vibrio injenensis*,
a recently described species only known from two strains isolated from human patients
in Korea (*Paek et al., 2017*) (Fig. 2D). The U.K. clone was 99% similar to the type strain
M12-1144[T] and carried 441 genes not present in the Korean strain. Both strains carry
the *rtx* toxin operon (Table S4). Only two ARGs, including tetracycline resistance *tet34*,
could be identified at a 75% similarity cut-off in the U.K. isolate (Table S5). Unlike the
Korean isolate, the U.K. strain appears to contain a toxin similar to the Zona occludens
toxin which is known to increase mammalian intestinal permeability (*Fasano et al., 1995*),
with highest similarity (74% amino acid similarity) to the fish pathogen *Vibrio
anguillarum* (*Castillo et al., 2017*). The isolation of this virulent clone is of particular
interest as *Vibrio* species have been identified as high risk emerging infectious pathogens
in Europe due to the effects of climate change (*Lindgren et al., 2012*).

The third clone *Pseudomonas aeruginosa* (Fig. 2F) isolated from seawater was found
to belong to Sequence Type 667, which is represented by four genome-sequenced human
pathogens. This clone carries an arsenal of virulence genes (228 at $\geq 90\%$ nt identity;
Table S6) including elastase (*Gi et al., 2014*) and Type II, III, IV and VI secretion
systems. This *Pseudomonas aeruginosa* clone also carries a variety of ARGs (46 at $\geq 90\%$ nt
identity; Table S7), including triclosan- and multidrug efflux pumps and beta-lactamases,
including *OXA50* conferring decreased susceptibility to ampicillin, ticarcillin,
moxalactam and meropenem and resistance to piperacillin-tazobactam and cephalotin
(*Girlich, Naas & Nordmann, 2004*).

The fourth clone from estuarine mud was identified as *Escherichia coli* belonging to
Phylogroup B2, specifically Sequence Type 3304, represented by three other isolates, from
a human patient, a Mountain brushtail possum and one unknown (Fig. 2H). This isolate
carries a range of virulence genes (Table S8), including *chuA*, *fyuA* and *vat* known to play a
role in uropathogenicity (*Müller, Stephan & Nüesch-Inderbinen, 2016*), *set1A* associated
with enteroaggregative *E. coli* (*Mohamed et al., 2007*) and *ibeA*, *OmpA* and *AslA* aiding
brain microvascular epithelial cell invasion, known from avian pathogenic- and neonatal
meningitis *E. coli* (*Wang et al., 2011*). This clone contains a range of ARGs, including

multidrug- and aminoglycoside efflux pumps, a class C *ampC* beta-lactamase conferring resistance to cephalosporins and *pmrE* implicated in polymyxin resistance (Table S9).

## CONCLUSION

Our study utilized the low-cost and ethically expedient *Galleria* infection model to directly measure the presence of pathogenic bacteria in environmental samples without any prior knowledge of identity. As expected, some samples with low FIB counts contained pathogenic bacteria and some samples with high FIB counts showed low *Galleria* mortality (Fig. S2). We note that of four pathogenic isolates, only one was a coliform and only two were gut-associated bacteria. Two out of the four isolates have not been reported from the UK before and potentially represent EIDs. This highlights the fact that transmission risk extends beyond "usual suspects" and includes environmental and largely uncharacterized strains. Our relatively simple methods can provide a basis for future studies to detect pathogenic bacteria in diverse environments, to ultimately elucidate their ecological drivers and estimate human infection risk.

### Funding
Will H. Gaze was supported by NERC grant NE/M011259/1. The funders had no role in study design, data collection and analysis, decision to publish, or preparation of the manuscript.

### Grant Disclosure
The following grant information was disclosed by the authors:
NERC: NE/M011259/1.

### Competing Interests
The authors declare that they have no competing interests.

### Author Contributions
- Rafael J. Hernandez performed the experiments, analyzed the data, contributed reagents/materials/analysis tools, authored or reviewed drafts of the paper, approved the final draft.
- Elze Hesse conceived and designed the experiments, analyzed the data, contributed reagents/materials/analysis tools, prepared figures and/or tables, authored or reviewed drafts of the paper, approved the final draft.
- Andrea J. Dowling analyzed the data, contributed reagents/materials/analysis tools, authored or reviewed drafts of the paper, approved the final draft.
- Nicola M. Coyle analyzed the data, contributed reagents/materials/analysis tools, prepared figures and/or tables, authored or reviewed drafts of the paper, approved the final draft.
- Edward J. Feil analyzed the data, contributed reagents/materials/analysis tools, authored or reviewed drafts of the paper, approved the final draft.

# PeerJ

- Will H. Gaze analyzed the data, contributed reagents/materials/analysis tools, authored or reviewed drafts of the paper, approved the final draft.
- Michiel Vos conceived and designed the experiments, performed the experiments, analyzed the data, contributed reagents/materials/analysis tools, authored or reviewed drafts of the paper, approved the final draft.

## DNA Deposition

The following information was supplied regarding the deposition of DNA sequences:

Trimmed reads and assemblies have been uploaded at the NCBI, accession: PRJNA473311.

## Data Availability

Using the wax moth larva *Galleria mellonella* infection model to detect emerging bacterial pathogens. Dryad Digital Repository DOI 10.5061/dryad.130q4qb.

## Supplemental Information

Supplemental information for this article can be found online at http://dx.doi.org/10.7717/peerj.6150#supplemental-information.

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
