# Peer review of "Using the wax moth larva Galleria mellonella infection model to detect emerging bacterial pathogens"

_PeerJ, doi:10.7717/peerj.6150_

## Round 0.1 · original submission · Minor Revisions

The reviewers were overall complimentary of your work. They have suggested some relatively minor revisions that hopefully you will consider making as appropriate.

·

Basic reporting

Clear, well-referenced and well structured presentation of fundamental science.

Fig S1: Please could the authors add labels for location key at the end of each line on the graphs, I’m finding it impossible to match most the lines to the boxplots on Fig S2 by colour vision alone, a lot of the shades are very similar.

Proteus clone: I thought it was very interesting that this clone turned the Galleria brown instead of black, could this be noted in the main text as well as in the SI? I don't know what it signifies, but it may be something that another reader recognises as diagnostic of a particular virulence mechanism or immune response.

Experimental design

Lines 120-125: I was interested in why the authors decided remove a single drop of haemocoel from the infection site, rather then homogenising the whole larva to sample bacteria from throughout the haemocoel. I guess there are arguments for and against both methods – the method used may fail to sample pathogens which have moved away from the infection site (as systemic infection can proceed rapidly in wax moth larvae), but will reduce sampling of “native” bacteria present in and on the larvae. As homogenisation of whole worms seems to be more typically used in the literature, please could the authors use a couple of sentences to justify their choice of this technique? I guess a lot depends on whether you expect infection to be caused by single pathogen strains present in the environmental samples, or whether high virulence is a function of the sample community. Fig S3 is broadly consistent with the former, which might justify the authors' choice? Also, could the authors specify whether the inclusion criterion for this experiment was melanisation at the infection site, or any melanisation at all?

Validity of the findings

Results are robust. Raw data is provided.

Additional comments

A small number of typos:

Line 73: “in laboratory” should be “in the laboratory”
Line 76: add plural “s” to “supernatant”
Line 82: erroneous . after “in”
Line 130: missing space after “37°C”
Line 188: Citations of Fig S3 and S4 should be Fig S2 and Fig S3, respectively.
Line 246: remove – after “environmental”

Reviewer 2 ·

Basic reporting

Clear and unambiguous, professional English used throughout.

Literature references, sufficient field background/context provided.

Fig 1 it is not sufficiently clear: label on the X-axis (presumably hours ?) is missed on all three graphs. Besides, an interruption on the scale on the second third of the X-axis help to make be more readable the left side of the graf.

Experimental design

Hernandez et al. here report an alternative model of potential use for the screening of environmental pathogens.
The idea is interesting and original.
nevekess due the higt varibilty of this model

Validity of the findings

The idea is interesting and original.
l nerveless due to the high variability of this model some concerns must be clarified.

as read out the mortality it is weak and not fully appropriate,
a scoring scale similar to doi: [10.3390/jof4030108] it wilòl improve the word and the data

Reviewer 3 ·

Basic reporting

The publication of Hernandez et al. describes potential use of Galleria mellonella infectious model to detect pathogens from water. The paper is written in clear unambiguous professional English. The “Introduction” clearly explains research area and provides enough background information which are supported by appropriate references. Another sections are properly structured as well. Figures are relevant to the content and authors also provides 3 figures, row data and 9 tables in supplemental material. I appreciate such amount of data in supplemental material and row data as well, however I would like to suggest authors to include figures from supplemental material to main paper, it would be more clear to reader for better understanding of methodology and results. The authors also should provide better and more detailed description of all supplemental figures.

Experimental design

Methods are well described and provides sufficient details and enough information to replicate. I have only those minor points:
Material and methods, line 77: Please write more information about M9 buffet. Please, specify composition of buffer or company which distribute it.
Material and methods, Lines 79 – 82. Extraction of bacteria from sediment samples: After twice spinning down of sediment samples (first: 3000 rpm, 2 min; second: 500 rpm, 15 min) you collected supernatant with bacteria. Are you sure that you separated all bacteria from sediment? Is it possible that some of bacteria remained in pellet with sediment particles after spinning of tubes?

Validity of the findings

Data is robust, statistically clear and well described. I have only one question: How did you analyze results from G. mellonella killing assay? Did you use any statistical program to compare survival of larvae after inoculation of different water samples or different CFU of bacteria?
Figure 1. Please, add description of X axes in G. mellonella survival assay. What does X axis represent (hours, or days of survival)? I also suggest to add curve for controls with 100 % of survival as well, it would be better for illustration of the virulence power of each bacteria sample or inoculated CFU.

Additional comments

The main point of this manuscript is to use invertebrate model organisms Galleria mellonella to detect emerging pathogens. Survival of G. mellonella larvae infected with samples from water and sediment was measured by killing assay. Water samples with the highest lethal impact on larvae has been further identified by another methods confirmed presence of emerging bacterial pathogens and identified their pathogenic potential. Results shown that in vivo model organisms - G. mellonella can be used for such a first screening of pathogens from environmental samples. This method represent easy, low cost and replicable novel approach of testing water from environment. I find this idea very interesting and I have doubt only with some minor points listed upper.

---

## Round 0.2 · accepted · Accept

Thank you for your efforts in addressing reviewer comments and revising your manuscript.

#